# The Remplissage Technique for Hill–Sachs Lesions in Competitive Athletes: A Narrative Review

**Michele Novi * and Simone Nicoletti**

Orthopaedic Unit—CESAT, San Pietro Igneo Hospital, Fucecchio, 50054 Florence, Italy
* Correspondence: miche.novi@gmail.com

**Abstract:** Anterior shoulder instability with Bankart lesion and associated posterior humeral head injury (Hill–Sachs) is common in athletes. Several treatments have been proposed for the management of the Hill–Sachs lesion, from bone grafts or rotation osteotomies to capsulotendinous interposition, such as remplissage. This procedure has been shown to be safe and effective in increasing glenohumeral stability. However, the correct indication concerning the bone defect and its effects in terms of range of motion and function, especially in highly demanding patients, is still debated. This narrative review aims to present the current state-of-the-art of the posterior capsulotenodesis in association with Bankart repair, for treating anterior shoulder instability in competitive athletes.

**Keywords:** shoulder instability; Hill–Sachs lesion; remplissage; Bankart repair; athletes

## 1. Introduction

Anterior shoulder dislocation is a common injury in the young population, with forty percent of shoulder dislocations occurring in patients 22 years old or younger [1]. Among the young population, competitive athletes reported a higher recurrence rate versus nonathletes [2,3].

The recurrence in shoulder instability is related to both anterior glenoid defect and the Hill–Sachs lesion, a grooved defect of the posterosuperior aspect of the humeral head. An HS injury is present in up to 70% of patients with one episode of shoulder dislocation and up to 90% of patients with recurrent instability [4].

Hill–Sachs (HS) lesion is a compression fracture caused by the impact of the anterior glenoid edge in the posterolateral aspect of the humeral head after an anterior dislocation [5].

Recurrent shoulder dislocation episodes can increase the humeral defect's size, leading to a higher risk of further instability.

From the original "engaging/non engaging" concept introduced by Burkarth and De Beer to the more recent concept of "on-track/off-track" lesions proposed by Di Giacomo et al., the relationship between the humeral defect and glenoid surface represents an essential factor for the recurrent instability and failure of isolated arthroscopic Bankart repair [6–8].

The leading surgical solutions proposed for glenohumeral instability with critical HS lesions varies from an open procedure such as the Bristow-Latarjet [9], osteochondral allograft for the bone defect [10] and rotational humeral osteotomy [11], to arthroscopic capsular-tendon interposition. The latter, described with the French word "remplissage," by Wolf et al. in 2004 [12] and Purchase et al. [13] in 2008, referred to the original procedure proposed by Connolly in 1972.

Whether more complications [14] burden the open procedures, the arthroscopic "remplissage" represents a less-invasive and effective technique, but only a few studies investigated its effect on the residual range of motion, functionality and complications. Some authors are concerned about a possible higher risk of recurrence rate and external rotation deficiency, especially in the collision and overhead athletes [15–17].

This narrative review aims to present a comprehensive summary of today's state-of-the-art of the posterior capsulotenodesis in association with Bankart repair for anterior shoulder instability and to understand how it impacts competitive athletes.

## 2. The Hill-Sachs Lesion

Hill–Sachs (HS) lesions were first described in 1890 by Broca and Hartman but formerly classified fifty years later by Hill and Sachs as a grooved compression fracture in the posterolateral aspect of the humeral head caused by the anterior glenoid edge after an anterior dislocation [5].

The HS lesion is a common finding in traumatic glenohumeral anterior instability. The prevalence of HSL is 65% to 67% after initial dislocation and 84% to 93% in recurrent dislocations [4,18].

Recurrent dislocations yield an increasing size of the HS lesion with a higher risk of further recurrence [19]. Severe defects are correlated with a higher risk of recurrence rate when treated with Bankart repair only, especially during abduction and external rotation movement of the arm [6,20].

Yamamoto and Itoi proposed the concept of the 'glenoid track' in a 3D CT-scan study, where they described the anatomical contact area between the posterior articular surface of the humeral head and the glenoid with the shoulder in external rotation and different degrees of abduction. When a HS lesion remains within this contact zone, it does not engage with the glenoid. However, when the HS is more medial, or when not entirely overlaying by the glenoid surface because associated with a bony-Bankart defect, HS engages, and the humeral head dislocates. The glenoid track is the distance between the medial margin of the contact area and the medial margin of the tuberosity; it is a value of 83% of the glenoid width when the arm was at 90° of abduction in live shoulders [21,22].

When a bony defect of the glenoid is present, the size of the resulting glenoid track is reduced because the measure of the glenoid defect width (called 'd') is subtracted from the 83% value (0.83D) to obtain the actual width of the glenoid track (0.83D−d). The width of the HS defect is as important as its location, as reported in the study of Kurokawa et al., where the authors distinguish between large-wide type off-track defects and narrow but medial off-track defects [23] (Figure 1).

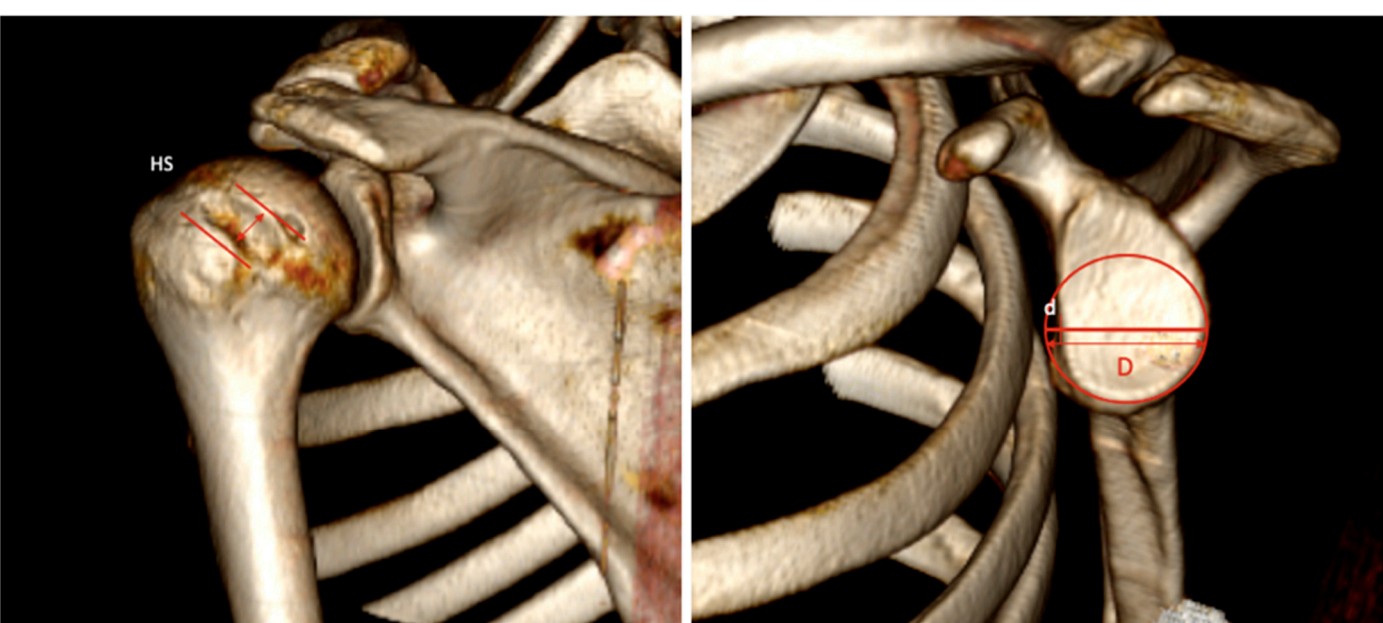

**Figure 1.** Glenoid Track GT: (0.83D—d). HS > GT: "Off-Track" Lesion. HS < GT: "On-Track" Lesion.

The evolution of the glenoid track concept was developed by Di Giacomo et al. [7] when they defined the HS lesion that engages as an 'off-track' HS, whereas the one that

does not engage was described as an 'on-track' HS. Consequently, to achieve shoulder stability, an off-track HS lesion has to be corrected into an on-track.

## 3. Sport-Related Injury Mechanism

Decision-making in shoulder instability in athletes depends on several factors, including age, injury pattern, and sport-related epidemiology. Owens et al. report the incidence in the population of 0.12 per 1000 athletes, with the highest rate in spring football (0.40 per 1000 athletes).

Most of the cases (68%) are caused by a collision with other athletes, the incidence rate is higher in male athletes (especially in spring football) than women, with a ratio of 2:1 [24,25].

Many epidemiological studies show how recurrence increases in the younger population. Moreover, Simonet and Cofield reported that young athletes had a recurrence rate of up to 90% (in collision sports) [3] versus 30% in young nonathletes [2].

*Time Lost from Sport*

The literature on time lost from sport and return to play after shoulder dislocation is limited.

An epidemiological study on professional rugby players conducted by Headley et al., reported that glenohumeral instability is the shoulder injury with the greatest amount of time lost from sport [26].

The amount of time loss from play after a shoulder dislocation varied by sport as reported by Owens et al.: 55% of the injuries stopped the athletes for less than 10 days, whereas 45% required at least 10 days or more to return to play. Football accounts for the majority of the days lost from play for men, whereas gymnastic for women athletes [24].

Buss et al. reported an average period of 10.2 days before return to play in almost 90% of the athletes who suffered from anterior dislocation [27]. The reported percentages of athletes that experienced a recurrent episode within the season vary from 30% [28] to 37% [27].

Another important aspect reported in the study by Headey is that shoulder dislocations occurred during the training sessions—especially during defensive training session—were significantly more severe (61 days) than those sustained during match play (27 days) [26].

The percentage of athletes who return to sport and full active duties after arthroscopic stabilization of shoulder instability is relatively high, usually in a mean period from 4 months to 7 months [29,30], however no studies, to our knowledge, report specific data related to remplissage.

## 4. Surgical Procedure

Several variations of posterior capsulotenodesis procedure are described, especially regarding the type of the anchors, number and position of the anchors, and type of the knots. However, the main steps we report are the same for all the procedures.

The patient is placed in the beach-chair position under general anesthesia with an interscalene nerve block. A posterior (P), anterosuperior (AS), and accessory posterolateral portal are used. The posterolateral called "remplissage portal" (RP), is centered over the Hill–Sachs lesion. Suture anchors, through the posterolateral portal, can be inserted perpendicular to the surface of the humeral bone defect.

*i.    Glenoid Preparation*

With the view from the posterior portal, the labrum and the inferior gleno-humeral ligament are mobilized with a Chisel dissector to be shifted superiorly and laterally. A temporary stay-suture for a proper traction is passed through the anterior labrum and capsule, then the anterior glenoid rim is gently debrided from 2 to 6 o'clock (referring to a right shoulder) with a shaver.

*ii.    Hill–Sachs Preparation*

Using a switch-and-stick, the camera is transferred to the AS portal. With the humeral head anteriorly translated, a spinal needle is introduced to localize the posterolateral portal directly perpendicular and central to the bone defect. The humeral defect is gently debrided with a shaver. One or two single-loaded soft-anchors are inserted (the author's preferred technique), respectively, one superior and one inferior and both adjacent to the medial margin of the defect. Solid anchors can be used as well, with similar biomechanical properties [31].

*iii.    Posterior Suture Passing*

The arthroscope is maintained in the anterosuperior portal. Sutures are managed from the remplissage portal. The suture threads are retrieved with a suture passer in a "V" shape through the capsule and the infraspinatus tendon (Figure 2), then retracted from the same portal and tagged.

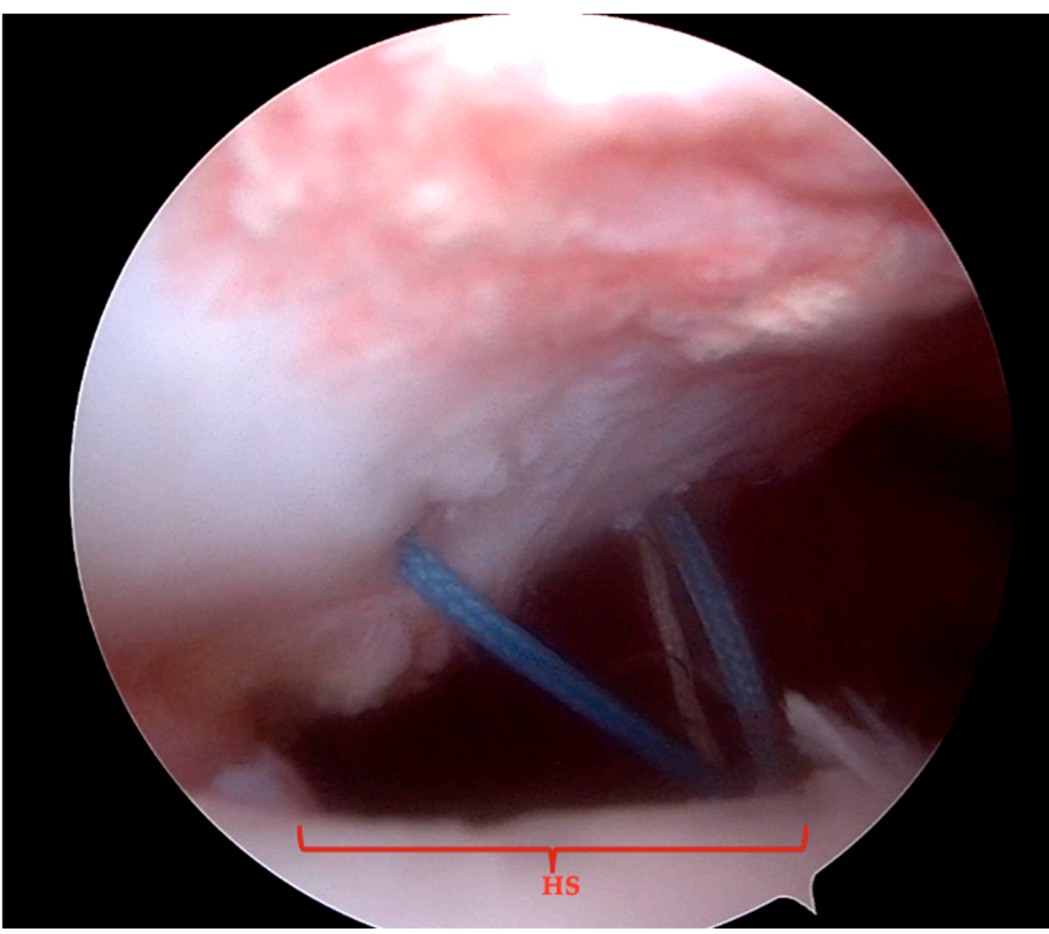

**Figure 2.** Suture anchor on the lateral edge of the HS lesion, with sutures passing through the posterior capsule for a "remplissage" procedure.

*iv.    Anterior capsuloplasty (Bankart repair) and secure the "remplissage-suture"*

At this point, Bankart repair of the anterior labrum is performed with an average of three or four single-loaded all-suture anchors. Once the Bankart suture is completed, the remplissage sutures are tightened to fill the HS defect. Different types of knots are described: single anchor with matters suture, two mattress sutures [32] or modified double-pulley suture [33] are usually performed.

A modified double-pulley technique provides two mattress sutures: one strand from each of the two anchors inserted in the Hill–Sachs defect are knotted together from outside

the arthroscopic portal. The two free strands are pulled together: by pulling the free strands, the "mattress" suture slides and secures the infraspinatus tendon and capsule over the HS defect (Figure 3). Once the two free strands have been tractioned tight, they are knotted down on the tendon with a knot-pusher in blind mode. The camera placed in the anterolateral portal is used to observe the complete adhesion of the tissue over the bone defect. The tension of the capulotenodesis is now tested with a probe [34].

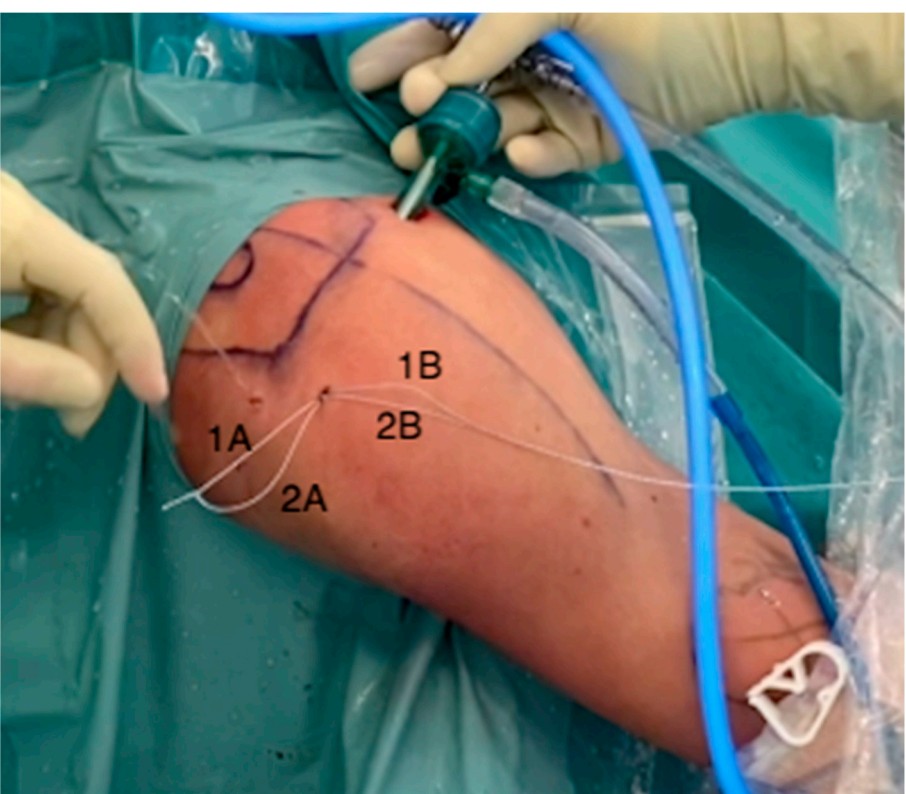

**Figure 3.** Modified double-pulley remplissage. One strand from the first anchor (1A) is tied up with a strand from the second anchor (2A). The two free ends of the suture (1B and 2B) are then pulled together from the same portal; 1A–2A "mattress" slides and reduces the infraspinatus tendon and capsule on the HS defect. The other two free ends are sutured with a knot pusher on the cuff as well.

*Effects of Surgical Technique on Range of Motion and Stiffness*

Limited studies in the literature have investigated the biomechanical effects of different surgical techniques in posterior capsulartenodesis.

In the cadaveric model described by Omi et al. [19] from the Mayo Clinic, the remplissage procedure for a large HSL caused significant restrictions in ROM, especially external rotation with arm in abduction and adduction. The authors recommend caution with this procedure in overhead athletes. Different results are reported by the biomechanical model of Argintar et al. [35], where HS remplissage had no statistically significant effect of on ROM or translation.

In their in vitro study, Giles et al. reported that the remplissage procedure is an effective procedure to prevent humeral engagement; however, it resulted in a rotation restriction and increased joint stiffness [36].

The remplissage is performed by suturing the posterior capsule and infraspinatus tendon into the Hill–Sachs defect. Different techniques of anchor implantation and soft tissues suture can be performed, with a possible different biomechanical response. An in vitro biomechanical study compared three remplissage techniques with different suture and anchor positioning and assessed their effects on joint stability and range of motion [17].

When the remplissage technique is performed as described initially by Purchase et al. [13], one or two anchors are placed in the center of the defect. Horizontal mattress sutures are

passed with an angle of around 90° through the posterior capsule and the infraspinatus (Figure 4A). A modification of the original technique involves the placement of the anchors into the rim of the subchondral bone in the Hill–Sachs defect, along the margin of the articular surface. The sutures are passed through the adjacent joint capsule and infraspinatus tendon (Figure 4B). Another proposed technique (Figure 4C) provides anchor placement as described in the first technique but the sutures are passed one centimeter more medial through the posterior joint capsule and infraspinatus tendon.

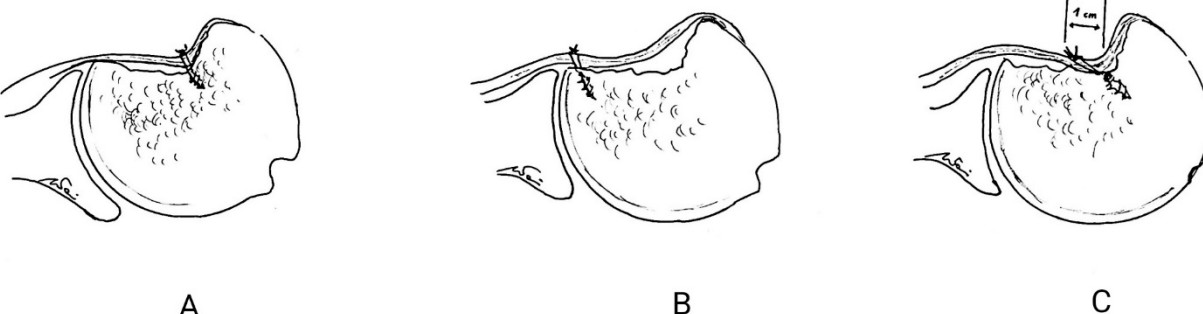

**A** **B** **C**

**Figure 4.** The remplissage ("to fill-in"/Fr.) is performed by suturing the posterior joint capsule and infraspinatus tendon into the Hill–Sachs defect. The suture anchor's position and the thread position may vary from (**A**) into the center of the Hill–Sachs defect, or (**B**) into the subchondral bone adjacent to the articular margin. (**C**) The anchor is positioned in the center of the defect with suture thread passed one centimeter medial through the posterior capsule and tendon.

All these three techniques significantly increase joint stiffness. However, they found that anchors placed in the valley of the defect with sutures passed medially resulted in significant joint stiffness. For minor HS defects (15%), only the third technique significantly reduced rotational ROM, with a mean decrease of 21° compared to Bankart repair alone.

For more significant HS defects (30%), this in vitro study reported that all remplissage techniques significantly reduced combined rotational ROM in adduction testing and a more significant reduction in rotational ROM for technique three (Figure 4C) compared with the other two techniques during abduction testing. These results should be considered, especially when dealing with instability in overhead athletes.

## 5. Outcomes

This paper reviews the anatomical and functional results after arthroscopic Hill–Sachs "remplissage" combined with Bankart repair in competitive athletes presenting with recurrent anterior glenohumeral instability associated with an isolated engaging humeral head bone defect.

A systematic review by Longo et al. on 26 studies with a total of 769 shoulders examined, compared the remplissage procedure, humeral osteochondral grafts, Weber osteotomy, and shoulder arthroplasty for the management of humeral bone defects. The authors reported a lower recurrence rate when the procedures are compared with Bankart repair alone and a lower rate of post-operative complications compared to open procedures [37].

It is reported that the overall recurrence of dislocation after a surgical treatment accounts for 6.5% of the shoulders. When a glenoid bone defect exists, this rate rises to 7.2%, a recurrence of 13.3 % in shoulders with humeral bony defect and 6.3 % in shoulders with both glenoid and humeral involvement [38].

Over the years, the growing interest in the arthroscopic remplissage technique has increased the number of clinical studies showing satisfactory and reliable results for treating glenohumeral instability in cases with significant and medial HS defects.

Decision-making on managing the humeral and glenoid defects in traumatic anterior glenohumeral instability remains uncertain, especially in highly demanding patients. There

is a lack of consensus on critical defect values and the exact percentage of bone loss leading to a higher risk of redislocation and the residual limitations in sports activities [38–40].

Most studies demonstrate that remplissage remains an excellent surgical option for engaging Hill–Sachs lesions in the presence of Bankart lesions with a limited glenoid bone loss, producing a failure rate from 6.5%33 to 11.8% [29]. However, Garcia et al. considered an unpredictable return to throwing sports, as only 50% of patients returned to sports such as baseball [29].

Assessing the functional outcome of remplissage on the range of motion, stiffness, and side symptoms are essential when, for example, an overhead athlete whose performance strictly depends on the extension of the external rotation and abduction. Nourissat et al. [16] proposed a level 2 prospective study where he compared two groups of patients with HS defect without glenoid bone deficiency: one treated by Bankart repair only, the other with Bankart repair and remplissage. No significant differences in the range of motion and recurrence rate were found between the two groups, confirming that remplissage is an effective and reliable technique. However, posterior residual pain was reported in one-third of the patients two years after the remplissage procedure [16]. This finding can be explained as incomplete tendinous healing or a posterior impingement of the articular side of the medialized rotator cuff. This irritative mechanism is likely to affect overhead elite athletes reducing the indication of the infraspinatus tenodesis versus the Latarjet procedure [41].

Infraspinatus tendon healing and strength recovery were assessed by ultrasound and clinical test by Merolla et al., reporting a satisfactory recovery of strength in the patient with a painful shoulder, with strength values near to the mean group value of patients enrolled in the study. They also reported that reduction in external and internal rotation does not significantly affect the quality of life, and infraspinatus strength recovery is satisfactory even compared with healthy subjects [42].

### 6. Conclusions

Arthroscopic remplissage for engaging Hill–Sachs defects is an effective and reliable technique to manage glenohumeral instability associated with Bankart and minor bony-Bankart defects, even in competitive athletes. It is associated with fewer complications compared to open procedures and decreases the recurrence rate when compared with Bankart repair alone. Clinical relevance of ROM limitations, especially in external rotation, is still debated. An accurate study of the injury pattern, especially for the medial and the larger HS lesions, and a correct position of the anchors are mandatory to improve joint stability and recurrence rate in competitive athletes. Return to sport has a high rate, but limited literature is present regarding the effects of remplissage combined with the Bankart procedure on the range of motion, especially in overhead athletes. Furthermore, level I and level II studies are necessary to investigate the impact of the Hill–Sachs remplissage in the sport-specific athletic gesture.

**Author Contributions:** All the authors equally contributed to the conceptualization of the manuscript. All authors have read and agreed to the published version of the manuscript.

**Funding:** This research received no external funding.

**Institutional Review Board Statement:** Not applicable.

**Conflicts of Interest:** The authors declare no conflict of interest.

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
