# Peer review of "The Remplissage Technique for Hill–Sachs Lesions in Competitive Athletes: A Narrative Review"

_2673-4036, doi:10.3390/osteology2040018_

Round 1

Reviewer 1 Report

Comments and Suggestions for Authors

General - This narrative review really does not provide any novel information regarding the remplissage technique and the analysis of the current literature is relatively basic. Quality of writing is poor and requires significant editing for appropriate grammar. For these reasons my recommendation would be against publication.

Page 2 - a figure demonstrating the calculations for on-track, off-track lesions would be useful

Figure 1 - Better arthroscopic images should be obtained showing a hil-sachs lesion pre and post remplissage

Figure 2 - Not clear what this is depicting 

Adding in a discussion of the pro's and cons of remplissage and bankart repair versus open latarjet in the management of subcrtical bone loss would be useful.

A brief discussion of other surgical treatment options for hil-sachs lesions would be useful, along with the pro's and con's of each option.

Numerous techniques exist to perform a remplissage, the author's describe their preferred however a brief discussion of other available options as well as the advantages and disadvantages of the various techniques would be useful.

Author Response

Thank you for your suggestion. Grammar revision has been edited. More reference has been edited. This study is not a systematic review on shoulder instability, but it would be a state of the art of the humeral posterior capsulotenodesis (aka remplissage) in sport athletes, so we tried to focus on that and even references are very selected in that way.

Ponit-by-point answer

Reviewer 1_ a figure demonstrating the calculations for on-track, off-track lesions would be useful

Answer: yes, provided

Figure 1: Better arthroscopic image is provided

Figure 2: changed. Integrated with a caption

thank you for reviewing our work.

Kind regards

Michele Novi

Reviewer 2 Report

Comments and Suggestions for Authors
  1. I would not describe this paper as a systematic review unless you apply Prisma guidelines.  It might be better described as a Surgical Technique and Review of the Literature. 
  2. There were a few locations where data was presented w/o citation.

3. The article requires spell check and extensive English language grammar modifications.

Line 17

Perhaps 'unequivocal' might be a more appropriate word here?

Line 19

Grammer.  Recommend you rework this sentence. ' The purpose is to present a systematic review demonstrating the current state of the art in terms of poserior capsultenodesis for the treatment of anterior shoulder instability for competitive athletes'.....or something similar to that.

Line 24

Grammer:  'a young population'

Line 24

Grammer:  'with forty percent of shoulder dislocations occuring'

Line 25

Grammer:  This isn't a complete sentence.  I think you are trying to say that anterior dislocations are typically seen in athletes.

Line 28

Grammer:  This sentence is not clear.  I think you are trying to say that HS lesions are associated with recurrence.   Is this status post surgery or  after non op treatment.

Grammer:  I recommend you use 'found' rather than 'find'

Line 31

Grammer:  I would recommend you state that HS lesions ARE a compression fracture rather than 'represent' lesions. 

Line 32

Syntax:  I would recommend a more complete discription  'Anterior glenoid edge impacting the humeral head'

Line 54

Grammar:  ‘Side Symptoms’ is not a common phrase….should you consider using the word ‘complications’?

Line 55

Grammar:  ‘Some authors ‘raise’ concern?  Recommend the use of recurrence rather than recurrency.

Line 70

Grammar:  Rather that saying ‘literature’ reports….I would simply state the findings and reference the article. 

Line 82

Grammar:  ‘Size of the resulting glenoid track is reduced’

Line 88

Grammar:  A bit too much for one sentence.  A paragraph should typically have 3 sentences, so you might break this explanation up. 

Line 94

Grammar:  ‘depends’.  The use of ‘of course’ is not needed here.

Line 102

Grammar:  recurrence ‘increases’

Line 110

Grammar:  A single sentence should not be a stand alone paragraph.  Consider combining the paragraph to follow.

Line 112

Grammar:  Consider ‘is the shoulder injury with the greatest amount of time’

Line 116

Clarification:  Consider ‘fewer than 10 days of time loss’

Line 121

Grammar:  Consider ‘sustained an’ rather than ‘suffered from’

Line 122

Grammar:  Percentage…..varies

Line 123

Grammar:  Run on sentence, consider breaking it up into 3 sentences. 

Line 127

Grammar:  Run on sentence, consider breaking up the first line

Line 129

Need citations here for the 118% and the 95.5%

Line 130

If you are going to present your numbers, you’ll need to state that in the objectives and then describe your methods and results.  If this is a ‘review’ of he literature you shouldn’t include your own data unless its published or presented as a study. 

Line 176

Grammar:  ‘are inserted with one superior and one inferior’

Line 197

Grammar:  ‘Tensioned’ rather than tractioned

Line 208

Grammar ‘scapular stabilization’

Line 222

Grammar:  Single sentence should not stand alone as a paragraph. 

Line 246

Clarification:  Significantly more?

Line 255

Grammar:  These rather than This

Line 270

Clarification:  This is a review and not a study. 

Line 277

This requires a citation  (for the recurrence rates noted)

Line 280

Clarification: You don’t really need to start the sentence with ‘however,’ but you could clean this sentence up  or break it into several sentences. 

Line 288

Grammar:  Try performed rather than proposed if the study has already occurred.

Line 328

Grammar:  I would not use the word ‘burdened’

Author Response

point by point answer to the reviewer.

Line 17 : Perhaps 'unequivocal' might be a more appropriate word here?

Answer: Sentence changed in “ it is still debated”

Line 19: Recommend you rework this sentence. ' The purpose is to present a systematic review demonstrating the current state of the art in terms of poserior capsultenodesis for the treatment of anterior shoulder instability for competitive athletes'.....or something similar to that.

Answer: sentence changed as suggested. Thank you

Line 24: grammer. Young population

Answer: done.

Line 25

Answer: changed. “Among young population, competitive athletes reported higher recurrence rate versus nonathlets”

Line 28:

Answer: thank you for the suggestion. Sentence changed: “Recurrence in shoulder instability is due both by the extent of anterior glenoid lesions and the Hill-Sachs lesion, a grooved defect of the posterosuperior aspect of the humeral head.”

Line 31:

Answer: ok , done.

Line 32:

Answer: Done

Line 54-55:

Answer: ok, modified.

Line 70:

Answer: “literature reports” removed.

Line 82:

Answer: ok, done.

Line 88: A bit too much for one sentence. A paragraph should typically have 3 sentences, so you might break this explanation up.

Answer: Yes, done

Line 94:

Answer: “of course” was removed

Line 102:

Answer: ok. Changed

Line 110:

Answer: corrected. “The amount of time loss from play after a shoulder dislocation varied by sport as reported by Owens et al.: fifty-five percent of the injuries stopped the athletes for less than 10 days, whereas 45% required at least 10 days or more to return to play.”

Line112:

Answer: corrected . “An epidemiological study on professional rugby players conducted by Headley et al, reported that glenohumeral instability is the shoulder injury with the greatest amount of time lost from sport.”

Line 116:

Answer:

Line 121:

Answer: ok, changed.

Line 122-127:

Answer: thank you. Modified.

Line 129: Need citations here for the 11.8% and the 95.5%

Answer: period changed and citations provided.

Line 130: If you are going to present your numbers, you’ll need to state that in the objectives and then describe your methods and results. If this is a ‘review’ of he literature you shouldn’t include your own data unless its published or presented as a study.

Answer: Thank you for the suggestion. Removed.

Line 176: “one superior and one inferior”

Answer: grammar correction.

Line 197: tensioned

Answer: done

Line 208: scapular stabilization

Answer: corrected.

Line 222: single sentence should not stand alone..

Answer: ok. Done

Line 246: clarification

Answer: I reported results from the study of Headey et al. (ref. n°26). I corrected the sentence specifying the severity of the trauma as number of days absent from sport.

Line 255: These rather than this.

Answer: ok, corrected

Line 270: this is a review, not a study

Answer: thank you. corrected

Line 277: citation

Answer: yes . provided

Line 280:

Answer: done

Line 288:

Answer: “performed” instead of “proposed”

Line 328:

Answer: sentence modified and word “burdened” has been removed

thank you for considering our work worthy for publication

kind regards

Michele Novi

Round 2

Reviewer 2 Report

Comments and Suggestions for Authors

Significant improvements in the paper.  Only one section requires minor editing:

"Discussion"  Should be renamed "outcomes" as the authors are not comparing and contrasting their personal study to prior work but are reviewing outcomes of other studies in this section.

In the first line the authors state 'This study reviews'....they should rather state 'this paper reviews' as this is not a study but a review paper

In terms of the discussion section the authors varied their paragraph style.  In the rest of the paper the authors indented the first line and did not have a line between paragraphs.  In this section the authors mixed style with some paragraphs not having an indention and being separated by a line.  

Author Response

1) "Discussion"  Should be renamed "outcomes" : 

Ok. Done

2) In the first line the authors state 'This study reviews'....they should rather state 'this paper reviews' as this is not a study but a review paper:

Ok. Sentence changed as suggested

3) In terms of the discussion section the authors varied their paragraph style:

Style corrected.thank you
